# Determination of Drug Use Behaviors and Related Reasons of Adult Patients Applying to Family Health Centers [note 1]

**DOI:** 10.3390/healthcare13222850

**Published:** 2025-11-10

**Authors:** Elif Deniz Şafak, Hilal Yüksel, Yusuf Kırış, Nimet Mısırlıoğlu Alper, Mümtaz M. Mazıcıoğlu

**Affiliations:** 1Department of Family Medicine, School of Medicine, Erciyes University, 38039 Kayseri, Türkiye; mumaz33@hotmail.com; 2Yenimahalle Şehit Ayhan Demirel Family Health Center, 06200 Ankara, Türkiye; doktor_hilal38@hotmail.com; 3Hayri Güldüoğlu Family Health Center, 38030 Kayseri, Türkiye; drysfkrs@hotmail.com; 4Fatma Yüksel İlbasmış Family Health Center, 38040 Kayseri, Türkiye; nm3417@hotmail.com

**Keywords:** drug adherence, rational drug use, health literacy, Morisky Medication Adherence Scale, Adult Health Literacy Scale

## Abstract

**Aim:** This study was conducted to determine the drug use behaviors of patients applying to primary healthcare centers and the factors affecting these behaviors. **Material and method:** This cross-sectional, analytic study included 913 individuals applying to family health centers for various reasons in Kayseri, Türkiye. All subjects completed a questionnaire that asked about sociodemographic characteristics and attitudes towards drug use via the Morisky Medication Adherence Scale and the Adult Health Literacy Scale. Data were analyzed using descriptive statistics. **Results:** A total of 913 individuals, comprising 288 (31.5%) men and 625 (68.5%) women, participated in this research. The average age of the participants was 41.79. Of the 913 subjects included, 23% reported that they would wait for recovery from a disease without any treatment attempt, while 53.5% reported that they visited a doctor, and 63.5% reported that they initially consulted a family health center. A total of 38.5% reported that they self-medicated without consulting a doctor. In addition, 79% of the subjects reported that they used medicine without a prescription. **Conclusions:** It was observed that age, gender, social insurance, educational status, level of health literacy, and presence of chronic diseases affect drug use behaviors. A weak, negative correlation was found between Morisky Medication Adherence scores and health literacy. Additionally, it was determined that only checking the expiration date before using a drug had an impact on drug adherence.

## 1. Introduction

Drug therapies constitute a foundational basis of modern healthcare, as they are pivotal for disease prevention, control, and treatment [1]. Rational drug use (RDU) is a health system policy that demands integrated effort from prescribers, pharmacists, nurses, and patients, and it aligns with the responsibilities of both health service providers and recipients [2]. Within this policy framework, states, public health institutions, and individual patients share accountability for achieving safe, effective, and cost-efficient medication use.

In many low- and middle-income countries, including Türkiye, irrational drug use imposes heavy burdens on health systems due to excessive drug prescription, self-medication, and dependence on imported pharmaceuticals. In particular, misusing and overusing antibiotics exacerbate antimicrobial resistance, constituting a global threat to health security [3].

To counter these trends, multiple countries have adopted RDU strategies. The goals typically include reducing adverse effects, preventing resistance, and optimizing therapeutic outcomes [4]. In Türkiye, the institution of the Family Medicine system in 2010 offered a structural lever for reinforcing rational prescription practices and continuity of care through assigned family physicians.

An essential determinant of therapeutic success is patient adherence. When physicians and patients align in expectations and when patients receive clear education about indications, dosage, and potential side effects, adherence improves [4]. Yet, in high-volume clinical settings, time constraints may hinder effective counseling, underscoring the role of primary care in supporting RDU.

In chronic disease contexts, studies have shown that higher rational drug use scores are positively correlated with better medication adherence and improved health perception [5,6]. Moreover, in primary care settings in Türkiye, patients often self-medicate before visiting clinicians, indicating gaps in community-level practices [7].

Considering this background, unlike previous studies, our study aimed to determine the drug use behaviors and the factors affecting these behaviors in patients applying to primary healthcare centers using a patient-centered approach.

## 2. Material and Method

Study design: This is a cross-sectional, analytical study aiming to investigate the drug use habits of patients applying to family health centers in Kayseri, Türkiye. In this study, all subjects answered a questionnaire on sociodemographic characteristics, drug use habits, adherence to usage instructions, prescription drug use, polypharmacy, and drug use habits, the 8-item Morisky Medication Adherence Scale (MMAS-8) and, the Adult Health Literacy Scale (AHLS). MMAS-8 and AHLS scores were obtained via a face-to-face interview method to determine drug adherence and health literacy levels, respectively. The sociodemographic survey was created by the researchers and administered by trained interviewers, and the answers were noted based on the participants’ self-reports.

The Morisky Medication Adherence Scale (MMAS-8): MMAS-8 is a self-reported scale that assesses drug use behaviors. It includes items that allow better assessment of factors causing treatment non-compliance. It was originally developed as a 4-item scale (MMAS-4) by Donald E. Morisky in 1986 [8], which was then modified to an 8-item scale in 2008. The minimum and maximum scores are 0 and 8, respectively. The total score is classified as follows: 0–3, poor adherence; 4–5, moderate adherence; and 6–8, good adherence. The Turkish validity and reliability of this scale was performed by Oğuzülgen et al. in 2014 [9].

The Adult Health Literacy Scale (AHLS): The Adult Health Literacy Scale was first developed by Sezer and Kadıoğlu in 2014 [10]. It includes 22 items about health knowledge and drug use and one figure about the localization of organs in the body, which aims to determine the competence of adult individuals in health literacy. The scale has 13 yes/no questions, 4 gap-filling questions, 4 multiple-choice questions, and 2 matching items. In the yes/no questions, answers indicating positive statements are rated as “1”, while answers indicating negative statements are rated as “0”. In the gap-filling items, correct answers are rated as “1”, while incorrect answers are rated as “0”. In the multiple-choice items, two or more correct answers are rated as “1”, while incorrect answers, as well as correct and incorrect answers together, are rated as “0”. In the matching items, answers with two or more correct matchings are rated as “1”, while other answers are rated as “0”. The total score ranges from 0 to 23 points. A higher total score indicates a higher health literacy level.

Study population and sampling: The study population consisted of adult patients who applied to family health centers in Kayseri province during a 3-month period. The minimum sample size was calculated as 913 subjects to sample 1% of the adult population in Kayseri province (913,000). At the time of the study, the adult population (>18 years of age) of two central districts (Melikgazi and Kocasinan) was 363,765 and 264,845, respectively. Therefore, we planned to recruit 363 and 264 subjects (88%) from Melikgazi and Kocasinan, respectively. The remaining subjects (12%) were to be recruited from the sub-provinces of Talas, Develi, Bünyan, and Hacılar.

Ethical approval and Study procedure: This study was conducted in accordance with the Declaration of Helsinki principles, and institutional ethical and legal permissions were obtained. Ethical approval for this study was obtained from the Faculty of Medicine Clinical Research Ethics Committee of Erciyes University (decision number 2015/367). Written informed consent was obtained from the patients. Administrative approval was obtained from the Kayseri Public Health Directorate.

Inclusion criteria: Individuals who were over 18 years of age, had been living in Kayseri for at least three years, signed the consent form prepared for the research, and completed all survey forms were eligible for the study.

Exclusion criteria: Individuals who applied to the family health center as guest patients, were under 18 years of age, did not sign the consent form, or filled out the survey forms incompletely were excluded from the study.

Statistical analysis: All statistical analyses were performed using R 2.14.0 (www.r-project.org (accessed on 25 January 2017)). The independent samples *t*-test and one-way ANOVA test were used to compare independent groups. Additionally, Pearson’s correlation analysis was used to evaluate relationships between demographic variables, the Morisky Medication Adherence Scale, and the Adult Health Literacy Scale. The Fisher–Freeman–Halton test was used to determine differences between the categorical variables. Univariate binary logistic regression was used to assess the Morisky Medication Adherence variable, demographic characteristics, and potential risk factors for drug use. The statistical significance level was set as *p* < 0.05.

## 3. Results

The mean age of the participants was 41.79 ± 15.06 years (min–max: 18–82). When age groups were assessed, it was observed that one-half of the participants were in the 18–39-year age group, while one-tenth were older individuals aged >65 years. One-fifth of the subjects had university degrees, while one-tenth were illiterate.

It was found that 20% of the subjects were single; 91.8% had social insurance; and 43.4% were on continuous drugs. When initial behavior in case of illness was evaluated, the majority preferred to seek professional medical care, most often through Family Health Centers, while a smaller proportion relied on home medications (Figure 1).

More than half of the participants reported using at least one drug without a prescription, most commonly painkillers and cough-cold preparations (Figure 2).

Adherence to the recommended drug dose and duration was significantly higher among male participants, whereas the tendency to request information about medications was higher among females (*p* < 0.05). Participants with university degrees requested less information, and the most common information sought concerned adverse effects (*p* < 0.05).

No significant gender difference was found in the rate of using or recommending drugs based on others’ advice; however, this behavior was more common among university graduates (*p* < 0.05). Only 11.7% of participants reported consulting a community member before visiting a doctor, and most of these were men. Alternative treatment use was uncommon (24.8%) and more frequent among men with higher education levels.

Nearly half of the participants (44.8%) had a chronic disease, predominantly hypertension, diabetes mellitus, asthma, hypothyroidism/goiter, and rheumatic disorders. Consistently, the most frequently used drug groups were antihypertensives, antidiabetics, thyroid drugs, asthma/COPD medications, and antirheumatic agents.

Table 1 presents the associations between the total mean scores of the Morisky Medication Adherence Scale and the Adult Health Literacy Scale, depending on age, gender, marital status, occupation, employment status, and social insurance.

There was a weak, negative correlation between Morisky Medication Adherence scores and health literacy, and age. In other words, health literacy decreased as adherence scores increased. Moreover, it was found that there was a weak, positive correlation between Morisky Medication Adherence scores and chronic diseases, with drug adherence increasing as the number of chronic diseases increased. In addition, a negative correlation was found between health literacy and age, the number of chronic diseases and the number of drugs used (Table 2). This means that the health literacy rate decreased as age, the number of chronic diseases, and the chronic drug users increased.

In the logistic regression analysis on factors affecting drug adherence, only checking expiry date had a significant effect on drug adherence (Table 3).

## 4. Discussion

While drug adherence is the individual responsibility of the patients, it is an important public health issue addressed worldwide since it is directly related to the vast majority of society and has the potential to cause physical and emotional harm to patients in addition to economic damage due to excessive use and disrupted drug use. Drug adherence manifests as a pattern of behavior resulting from sociodemographic characteristics, education level, and communication with the environment.

In our study, it was found that 68.5% of the subjects were women, and more than one-third had primary school degrees; these findings are in agreement with the literature [11,12]. The reasons may be because more women, who are generally housewives, come to primary care compared to men and the majority of these women have primary school degrees.

It was also found that the proportions of participants who were married and those with chronic drug use were 2-fold higher than those reported in previous studies [11,12,13]. The higher proportions in our study may be due to higher rates of obesity and diabetes mellitus in the Central Anatolian region compared to other regions of Türkiye and polydrug use for treatment.

In our study, when the first actions taken during illness were analyzed, it was found that 53.4% of the subjects preferred to consult a doctor, indicating a higher rate than that reported in the literature [14,15]. Of the subjects who preferred to visit a doctor, 63.5% reported that they applied to a family health center first. This may be because the family practice system was implemented before our study and thus patients can readily access their registered doctor. The proportion of participants who visited a doctor first in case of sickness was higher among male subjects, which is in agreement with the literature [15].

In our study, it was seen that a significantly higher proportion of subjects with university degrees initially preferred not to take any action in case of getting sick while the rate of self-treatment was significantly lower. Possessing a high education level, awareness about viral infections, and an idea about treatment can explain the above-mentioned behaviors.

In our study, it was observed that 26.6% of the subjects did not take drugs prescribed by a pharmacy as the drugs prescribed were already available at home. Similarly, in a previous study involving 300 patients applying to family health centers, it was found that 15.7% of the participants did not take drugs prescribed by a pharmacy as the majority of the drugs prescribed were already available at home, indicating a higher rate of drug stocks at home [13]. This may be a result of the fact that in Türkiye, most drugs except antibiotics can be easily purchased from pharmacies without a prescription [16].

For rational drug use, it is a requisite to use drugs for the period recommended by the clinician. However, drugs are often used for a shorter time than recommended by clinicians, which facilitates the development of drug resistance [16].

In our study, it was found that more than two-thirds of the subjects requested information about the drugs they were using, particularly regarding adverse effects, the method of administration, and the effects of the drug. Similarly, it has been reported that patients mostly request information about adverse effects and the duration of treatment [14,17].

According to the literature, patients expect to be informed about their disease, the treatment duration, the adverse effects of the drug they use, when they can stop using the drug, and whether they will have follow-up visits by a clinician. This issue is also addressed in medical deontology and regulations on patient rights [18]. It has been reported in the literature that the request for information about treatment increases with increasing education level; in contrast, in our study, it was found that the proportion of patients requesting information from clinicians was significantly higher among female subjects and those without university degrees [13,17]. This may be a result of the overwhelming information about health issues available on social media platforms. Increasing awareness may have led to patients questioning the use, effects, and side effects of the drug used.

Drug use without a prescription is another important problem related to rational drug use. Although it is a legal obligation that drugs be sold only by prescription according to Act 1262 implemented in 1928 in Türkiye, it is often neglected [19]. Moreover, although many drugs in Türkiye are labeled as “Available only with prescription”, over-the-counter drug use can stem from the purchase of medicines without a prescription, especially by people without social insurance [20]. In our study, when participants were asked whether they buy drugs without counseling a doctor/prescription, 19.7% responded “yes”, while 18.8% responded “sometimes”. In the literature, the reported rate of buying drugs without consulting a doctor is higher compared to our study [15]. This result is possibly due to the prohibition of purchasing antibiotics from pharmacies without a doctor’s prescription at the time of data collection.

In our study, it was observed that painkillers (79.0%) were the most commonly used drugs without a prescription, followed by cough and cold medications (37.8%), antibiotics, and vitamin preparations. It was found that the rate of non-prescribed drug use, mostly vitamin preparations, was significantly higher among subjects without university degrees. Interestingly, the rate of painkiller use without a prescription was higher among subjects with university degrees. In previous studies from Türkiye, it was found that antibiotics, painkillers, anti-rheumatoid agents, and cold and cough agents were the most commonly used drugs without medical consultation [11,21,22]. Similarly, analgesic drugs and antibiotics are among the most frequently used non-prescription drugs according to the global literature [23,24,25].

Checking expiry dates before drug use and reading patient leaflets are considered two major drivers of drug use. When these factors were assessed in our study, it was found that more than 80% of the subjects responded “yes”. It has been reported in the literature that the rates of checking the expiration date and reading patient leaflets before drug use are high, similar to our study. It has been suggested the rates are significantly higher in subjects with secondary school or college degrees and in women [7,14,15,26,27]. This may be a result of the excessive discussion of health issues on social media and television programs, and the overexposure of people, especially housewives, to these programs.

In our study, similar to the literature, the rate of good adherence with drug was found to be low at 16.7% according to Morisky Medication Adherence Scale scores [9,28,29]. When the factors affecting participants’ drug adherence were evaluated, it was found that simply checking the expiration dates of drugs increased drug adherence by 1.654 times (95.0% CI: 1.011–2.708).

Non-compliance with treatment is one of the common problems faced in chronic disease management. It is known that non-compliance with treatment leads to increases in hospitalization, morbidity, premature death, and healthcare expenses. In the literature, it has been reported that the adherence rate can be as high as 50% for different chronic diseases and treatment modalities. When we reviewed studies assessing drug adherence in subjects with chronic diseases showing high incidence such as hypertension, asthma/chronic obstructive pulmonary disease, and in those with psychotic diseases requiring hospitalization, it was found that rates of drug adherence ranged from 40% to 60% in these studies and that rate of good drug adherence was about 20% in agreement with our study [9,28,29]. We believe that the effective use of the Morisky scale, which can be used as a tool to identify non-compliance and guide the interventions required, will contribute to chronic disease management.

The World Health Organization considers that ensuring drug adherence as well as health literacy has a key role in improving health. Health literacy is the key to ensuring the community access to health and healthcare, accurately perceiving messages given for promoting and improving health, concluding messages communicated, running communication channels to carry such information in all segments of the community, and enabling resort to healthcare providers at accurate place with accurate timing and frequency [30].

In our study, a weak, negative correlation was found between Morisky Medication Adherence scores and health literacy.

Studies in the literature indicate significant differences in health literacy, even among well-educated groups such as university students [31,32]. A high level of formal education does not automatically translate to high health literacy, as individuals may lack the critical skills needed to effectively evaluate and apply health information. High health literacy does not guarantee high medication adherence. Indeed, the correlation analysis in our study also revealed this result. Similarly to the paradox in our study, a negative correlation was reported between health literacy and adherence to medication treatment in the study conducted by Persell et al. [32]. Therefore, health communication strategies, both in universities and in the broader community, should prioritize improving health literacy.

### 4.1. Study Limitations

The answers given to our survey questions asking about drug adherence and drug use can be misleading since they are subjective and were not extracted from an available database or obtained from home visits.This study did not include follow-up or re-testing due to the cross-sectional nature. Thus, there may be an increased margin of error.A lack of individual interviews to eliminate factors that may affect survey answers, such as participants’ low education level, incorrect or incomplete information about the disease, or a lack of awareness of the seriousness of the disease, may have negatively affected the study results.

### 4.2. Study Strengths

The study population was selected to include 1% of the general population in Kayseri province and its sub-provinces and recruited from 25 family health centers, representing a wide spectrum of socioeconomic levels.The survey was conducted by the same three researchers using face-to-face interviews.The Adult Health Literacy Scale, which assesses the level of health literacy thought to affect medication adherence behavior, and the Morisky Medication Adherence Scale, a self-report measure to determine medication adherence levels among patients with chronic diseases, were used simultaneously.Unlike previous studies, this is the first study to reveal the drug use behaviors and health literacy levels of patients in Kayseri, Türkiye, with a patient-centered approach focusing on individual and social behaviors.

## 5. Conclusions

This study reveals that age, gender, social security, education level, health literacy level, and the presence of chronic disease influence medication use habits. The results of our study are important because they demonstrate the importance of patient- and community-level factors, such as clinicians’ knowledge of medications, their approach and provision of information to patients, and patients’ sociodemographic characteristics and social environment, in determining behaviors and attitudes toward medication use. These findings can serve as a guide for future research in this area.

## Figures and Tables

**Figure 1 healthcare-13-02850-f001:**
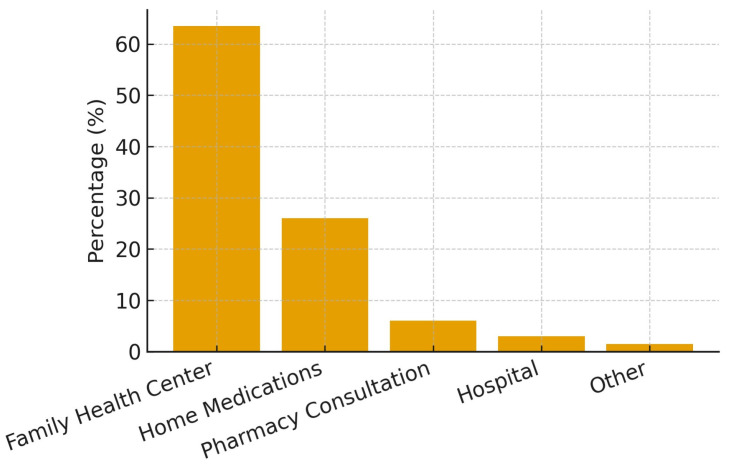
Initial health-seeking behaviors of participants when ill.

**Figure 2 healthcare-13-02850-f002:**
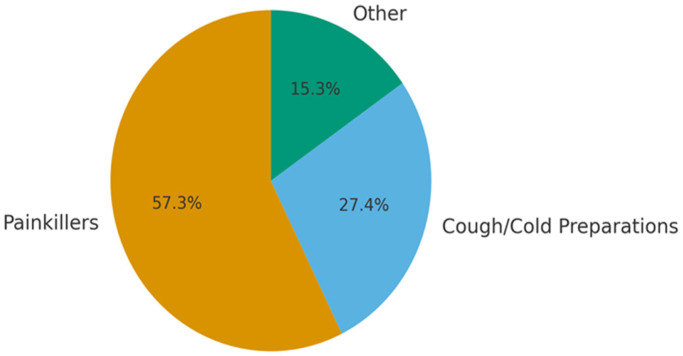
Distribution of drugs used without prescription among participants.

**Table 1 healthcare-13-02850-t001:** The relationship between Morisky Medication Adherence and Health Literacy in terms of sociodemographic characteristics.

		Morisky Score			Health Literacy Score		
		n	X (SD)	*p*	n	X (SD)	*p*
Gender	Male	288	4.49 (1.97)	0.054	288	11.81 (3.44)	**0.001 ***
	Female	625	4.94 (2.25)		625	10.98 (3.70)	
Age (year)	18–39	451	524 (2.45)	**0.038 ***	451	11.66 (3.54)	**0.001 ***
	40–64	364	4.57 (2.03)		364	11.33 (3.52)	
	>65	81	4.82 (2.14)		81	8.96 (3.75)	
Marital Status	Maried	752	4.82 (2.12)	0.911	752	11.35 (3.60)	**0.012 ***
	Single	112	4.48 (2.69)		112	11.19 (3.67)	
	Divorced	16	4.75 (2.60)		16	10.38 (3.63)	
	Widow	33	4.84 (2.29)		33	9.30 (3.92)	
Occupation	Housewife	471	4.98 (2.22)	0.069	471	10.40 (3.55)	**0.001 ***
	Officer	133	4.15 (1.95)		133	13.90 (2.73)	
	Laborer	115	5.02 (2.21)		115	10.72 (3.25)	
	Tradesman	82	4.50 (1.93)		82	11.44 (3.40)	
	Other	112	4.88 (2.27)		112	12.01 (3.84)	
Curent empyloyment status	Unemployd/ housewife Active worker Retired	514 303 89	5.02 (2.21) 4.72 (2.28) 4.32 (1.81	0.053	514 303 83	10.59 (3.61) 12.27 (3.54) 11.66 (3.15)	**0.001 ***
Social Insurance	No 4.75	28	4.75 (1.26)	0.739	28	10.29 (4.40)	**0.006 ***
	Social insurance agency	836	4.78 (2.16)		836	11.35 (3.56)	
	Health Card for uninsured people	29	4.80 (2.74)		29	9.21 (4.87)	
	Private insurance	8	6.33 (3.06)		8	9.75 (2.55)	
	Other	10	5.67 (2.08)		10	12.60 (1.58)	
Income level	$0–395	327	4.94 (2.25)	0.244	327	9.71 (3.51)	**0.001***
	$40–99	376	4.72 (2.09)		376	12.13 (3.14)	
	>$100	89	4.30 (2.05)		89	13.76 (3.28)	
Perception of income level	Well	175	4.76 (2.04)	0.408	175	11.87 (3.60)	**0.001 ***
	Middle	410	4.66 (2.22)		410	11.57 (3.58)	
	Poor	310	4.98 (2.18)		310	10.46 (3.59)	
Staff relatives	No	666	4.78 (2.22)	0.841	666	10.92 (3.67)	**0.001***
	Yes	247	4.83 (2.08)		247	12.11 (3.41)	
Chronic drug use	No	517	4.77 (2.28)	0.927	517	11.66 (3.44)	**0.001***
	Yes	396	4.80 (2.16)		396	10.70 (3.82)	
Chronic disease	No	50	4.96 (2.37)	0.573	504	11.49 (3.50)	**0.024***
	Yes	346	4.77 (2.14)		409	10.94 (3.78)	
Use of traditional and complementary medicine	No	105	5.17 (2.19)	**0.040 ***	226	11.97 (3.58)	**0.001***
	Yes	291	4.66 (2.17)		687	11.00 (3.63)	

* *p* < 0.05. Independent Samples *t*-test was used for two groups. One-way ANOVA was used for more than two groups. Significant *p* values are given in bold.

**Table 2 healthcare-13-02850-t002:** The relationship between Morisky Medication Adherence and Health Literacy and the chronic drug use, age, and chronic diseases.

	n	r	*p*
Morisky Score & Chronic drug use	396	0.005	0.917
Morisky Score & Age	393	−0.106	**0.035 ***
Morisky Score & Health Literacy Score	396	−0.208	**0.001 ***
Morisky Score & Number of Chronic Disease	341	0.134	**0.013 ***
Health Literacy Score & Chronic drug use	913	−0.194	**0.001 ***
Health Literacy Score & Age	896	−0.181	**0.001 ***
Health Literacy Score & Number of chronic diseases	402	−0.260	**0.001 ***

r: Pearson’s correlation coefficient. * *p* < 0.05. Significant *p* values are given in bold.

**Table 3 healthcare-13-02850-t003:** Logistic regression analysis for Morisky Medication Adherence variable.

	OR	95.0% CI		*p*
Checking expiry dates before drug use	1.654	1.011	2.708	0.028
Using medication with advice	1.179	0.618	2.250	0.032
Advising someone to use medication	1.401	0.761	2.580	0.041
Getting help while use medication	2.440	0.912	6.530	0.038
Gender	1.503	0.959	2.357	0.025
Using non-prescription medication	1.511	0.879	2.598	0.043
Consulting a doctor				

*p* < 0.05. OR: Odds ratio. CI: Confidence interval.

## Data Availability

The dataset used and analyzed in this study is available from the corresponding author. The data are not publicly available due to ethical restrictions.

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
