# Peer review of "Determination of Drug Use Behaviors and Related Reasons of Adult Patients Applying to Family Health Centers [Author-notes fn1-healthcare-13-02850]"

_healthcare, 2025, doi:10.3390/healthcare13222850_

Round 1

Reviewer 1 Report

Comments and Suggestions for Authors
  1. The manuscript requires English editing for the correct use of words and grammar.
  2. The introduction part lacks coherence within paragraphs and needs some editing.
  3. The author has not cited previous literature on medication adherence and drug use behaviors.
  4. How does the author differentiate between drug use behavior and medication adherence? Are they two different concepts? Kindly elaborate.
  5. A lot of information is provided in the study design. It's better to give separate headings for ethical approval and study procedure and add relevant material under these headings.
  6. The study population heading also contains information on the sample size calculation. It's better to modify the heading to ‘study population and sampling’.
  7. In the results reported in lines 137-139, the author did not report any findings on antibiotics or antipsychotics without prescription use. Was any data collected on their misuse?  
  8. The tables are added as images. Kindly add tables properly
  9. I could not see the results for the Tukey test and univariate binary logistic regression. No p-values available for univariate binary logistic regression.
  10. The conclusion should be written under a separate heading.
  11. The lines 327-329 should not be part of the conclusion and can be added to the last paragraph of the introduction section.

Author Response

Dear Reviewer, Thank you very much for taking the time to review this manuscript. Respond for the manuscript “Submission ID: healthcare-3890681” specific recommendations and comments are listed below. Each change is made point by point and changes corrections and moved parts in the main manuscript. Thanks for your interest. Best regards Respond 1#. The manuscript requires English editing for the correct use of words and grammar. The manuscript had previously undergone language editing by the Proofreading & Editing Office of the Dean for Research at Erciyes University. However, based on the suggestions, it was re-evaluated by MDPI's professional Language and Figure Editing unit for correct grammar and vocabulary. Suggested corrections are indicated in the text. 2#. The introduction part lacks coherence within paragraphs and needs some editing. The introduction section has been edited to ensure consistency between paragraphs as suggested. 3#. The author has not cited previous literature on medication adherence and drug use behaviors. In line with the recommendations, excerpts from previous literature on medication adherence and drug use behaviors have been included in the text. 4#. How does the author differentiate between drug use behavior and medication adherence? Are they two different concepts? Kindly elaborate. Drug use behavior includes factors such as whether the medication was obtained by prescription or over-the-counter, who the person consulted when they first became ill, and whose advice led to the medication being started. However, medication adherence refers to whether the person takes the prescribed medication regularly, as directed, at the appropriate dose, and for the appropriate duration. 5#. A lot of information is provided in the study design. It's better to give separate headings for ethical approval and study procedure and add relevant material under these headings. The study design is divided into subheading: "Ethical approval and Study procedure." Relevant materials are grouped under this subheading. 6#. The study population heading also contains information on the sample size calculation. It's better to modify the heading to ‘study population and sampling’ The study population heading has been changed to "study population and sampling." 7#. In the results reported in lines 137-139, the author did not report any findings on antibiotics or antipsychotics without prescription use. Was any data collected on their misuse? Lines 137-139 of the study revealed that 61.4% of participants obtained and used medications without consulting a doctor or without a prescription, with painkillers being the most frequently used. Because the study design did not consider a hypothesis regarding medication abuse, no data were collected for this purpose. 8#. The tables are added as images. Kindly add tables properly. Thank you for this justified warning. The tables have been rearranged. 9#. I could not see the results for the Tukey test and univariate binary logistic regression. No p-values available for univariate binary logistic regression. Because the memory in which we stored our study data was malfunctioning, Tukey analysis could not be performed, and p values could not be obtained in the univariate binary logistic regression. Therefore, the term Tukey test was removed from the statistics section. 10#. The conclusion should be written under a separate heading. The conclusion section was written under a separate heading as suggested. 11#. The lines 327-329 should not be part of the conclusion and can be added to the last paragraph of the introduction section. Thank you for pointing this out. I/we agree with this comment. Therefore, lines 327-329 and the study's strengths have been reorganized to eliminate ambiguity and highlight its strengths. Additionally, the final paragraph of the introduction has been reorganized for clarity, based on these suggestions. All corrections are highlighted in yellow within the main text. Thank you very much for your contributions.

Reviewer 2 Report

Comments and Suggestions for Authors

This is a solid paper with an very interesting subject. I congratulate the authors for conducting this study. My suggestions:

  • there is a lot of focus on Turkihs literature on this subject? Are there any international examples? I have a feeling that that would emphasise the strenght of this paper
  • could ypu please clarify the interpretation of the negative correlation between health literacy and adherence, as this finding appears counterintuitive
  • do you have more practical recommendations for healthcare providers and policymakers based on the findings of the study?

Author Response

Dear Reviewer, Thank you very much for taking the time to review this manuscript. Respond for the manuscript “Submission ID: healthcare-3890681” specific recommendations and comments are listed below. Each change is made point by point and changes corrections and moved parts in the main manuscript. Thanks for your interest. Best regards Respond 1#. This is a solid paper with an very interesting subject. I congratulate the authors for conducting this study. My suggestions: there is a lot of focus on Turkihs literature on this subject? Are there any international examples? I have a feeling that that would emphasise the strenght of this paper. Could you please clarify the interpretation of the negative correlation between health literacy and adherence, as this finding appears counterintuitive. Do you have more practical recommendations for healthcare providers and policymakers based on the findings of the study? Thank you very much for your kind comments and constructive criticism. These suggestions are really important to increase the impact of our article. In the Turkish literature, this topic has been addressed from a physician-centered perspective and evaluated in terms of rational drug use or health literacy. However, our study is the first to examine patients' medication use behaviors and health literacy levels with a patient-centered approach focusing on individual and societal behaviors. Furthermore, the study population was selected to represent a broad range of socioeconomic backgrounds, representing 1% of the general population. This is important to emphasize the importance of patient and societal factors, such as patients' sociodemographic characteristics and social environment, in determining behaviors and attitudes toward medication use. Recommendations for healthcare providers and policy makers based on the findings of our study are aded in the discussion and conclusion section. To this end, national and international literature was reviewed based on your suggestion, and recent studies that would enhance the strength of the article were added to the main text. All corrections are highlighted in yellow within the main text. Thank you very much for your contributions.

Reviewer 3 Report

Comments and Suggestions for Authors
  1. Originality / Novelty

While the topic of rational drug use and medication adherence is relevant, the novelty is somewhat limited since similar studies have been conducted in other regions and countries.

Suggestion: Emphasize more clearly what makes this study unique (e.g., the specific sociocultural or policy context of Kayseri, or comparison with previous national data).

  1. Significance of Content

The findings are significant at a local and policy level, but the discussion does not sufficiently highlight the broader implications.

Suggestion: Expand on how your findings could inform national health policy in Türkiye or be relevant for other countries with similar healthcare systems.

  1. Quality of Presentation

The manuscript is generally clear, but the Results section is lengthy and text-heavy. Important findings are sometimes buried in descriptive detail.

Suggestion: Consider reorganizing the Results to focus on key outcomes. Use figures or charts to illustrate major associations (e.g., relationship between health literacy and adherence).

  1. Scientific Soundness

The methodology is appropriate, but the analysis remains somewhat limited. For example, logistic regression identified only one significant factor (checking expiry dates), which seems surprising.

Suggestion: Consider including more comprehensive multivariate analyses to explore combined effects of sociodemographic variables. Discuss possible reasons why only expiry date checking emerged as significant, and whether this reflects limitations of the dataset.

  1. Interest to Readers

The study will be of interest to readers in primary care and public health, but the broader appeal is constrained by the localized scope.

Suggestion: Frame your findings in a way that emphasizes lessons for international readers—such as highlighting how health literacy influences medication adherence universally.

  1. Conclusions and Implications

The conclusions summarize the findings accurately but do not fully address their implications for clinical practice or public health policy.

Suggestion: Strengthen the conclusion by discussing practical recommendations for clinicians, family health centers, and policymakers.

  1. Figures and Tables

The tables are informative but dense. The absence of figures weakens the visual presentation of results.

Suggestion: Add simple figures (e.g., bar charts or correlation plots) to visually support key findings, making the paper more engaging for readers.

  1. Ethical and Methodological Limitations

You mention limitations (self-report, cross-sectional design), but these are not thoroughly discussed in terms of how they may have influenced the results.

Suggestion: Provide a deeper reflection on potential biases (recall bias, social desirability bias) and their impact on interpretation of findings.

Author Response

Dear Reviewer, Thank you very much for taking the time to review this manuscript. Respond for the manuscript “Submission ID: healthcare-3890681” specific recommendations and comments are listed below. Each change is made point by point and changes corrections and moved parts in the main manuscript. Thanks for your interest. Best regards Respond 1#. Originality / Novelty While the topic of rational drug use and medication adherence is relevant, the novelty is somewhat limited since similar studies have been conducted in other regions and countries. Suggestion: Emphasize more clearly what makes this study unique (e.g., the specific sociocultural or policy context of Kayseri, or comparison with previous national data). Thank you for this justified reminder. Unlike previous studies, this is the first to explore patients' medication use behaviors and health literacy levels using a patient-centered approach focusing on individual and societal behaviors. Furthermore, the study population was selected to represent a broad socioeconomic background representing 1% of the general population. This is important to emphasize the importance of patient- and societal-level factors, such as patients' sociodemographic characteristics and social environment, in determining behaviors and attitudes toward medication use. This distinctive aspect of our study has been highlighted and rewritten, as suggested, in the introduction, study strengths, and conclusion sections. 2#. Significance of Content The findings are significant at a local and policy level, but the discussion does not sufficiently highlight the broader implications. Suggestion: Expand on how your findings could inform national health policy in Türkiye or be relevant for other countries with similar healthcare systems. Broader implications, including the implications of the study results for health policies in Türkiye and countries with similar healthcare systems, were added to the discussion section as suggested. 3#. Quality of Presentation The manuscript is generally clear, but the Results section is lengthy and text-heavy. Important findings are sometimes buried in descriptive detail. Suggestion: Consider reorganizing the Results to focus on key outcomes. Use figures or charts to illustrate major associations (e.g., relationship between health literacy and adherence). The results have been condensed and reorganized to focus on key outcomes. Significant relationships were illustrated graphically. 4#. Scientific Soundness The methodology is appropriate, but the analysis remains somewhat limited. For example, logistic regression identified only one significant factor (checking expiry dates), which seems surprising. Suggestion: Consider including more comprehensive multivariate analyses to explore combined effects of sociodemographic variables. Discuss possible reasons why only expiry date checking emerged as significant, and whether this reflects limitations of the dataset. Thank you for this constructive suggestion. Medication use includes factors such as whether the medication was obtained by prescription or over-the-counter, who the person consulted when they first became ill, and whose advice led to the medication being started. However, medication adherence refers to whether the person takes the prescribed medication regularly, as directed, at the appropriate dose, and for the appropriate duration. There is no doubt that the addition of logistic regression and further analysis as suggested will make our study stronger. However, the memory in which we stored our study data did not work and unfortunately the recommended additional analyzes could not be added. 5#. Interest to Readers The study will be of interest to readers in primary care and public health, but the broader appeal is constrained by the localized scope. Suggestion: Frame your findings in a way that emphasizes lessons for international readers—such as highlighting how health literacy influences medication adherence universally. The study's findings were highlighted to attract the attention of international readers, with topics such as how health literacy universally influences medication adherence and the universal impact of medication use habits on healthcare. 6#. Conclusions and Implications The conclusions summarize the findings accurately but do not fully address their implications for clinical practice or public health policy. Suggestion: Strengthen the conclusion by discussing practical recommendations for clinicians, family health centers, and policymakers. Implications of the findings for clinical practice and public health, and recommendations for clinicians, family health centers, and policy makers were added to the conclusion section as requested. 7#. Figures and Tables The tables are informative but dense. The absence of figures weakens the visual presentation of results. Suggestion: Add simple figures (e.g., bar charts or correlation plots) to visually support key findings, making the paper more engaging for readers. Thank you for pointing this out. I/We agree with this comment. Therefore, the main findings were organized and supported by graphics. 8#. Ethical and Methodological Limitations You mention limitations (self-report, cross-sectional design), but these are not thoroughly discussed in terms of how they may have influenced the results. Suggestion: Provide a deeper reflection on potential biases (recall bias, social desirability bias) and their impact on interpretation of findings. How potential biases and methodological limitations may have influenced the results is discussed, and the findings are presented from this perspective in the main text and study limitations section. All corrections are highlighted in yellow within the main text. Thank you very much for your contributions.

Round 2

Reviewer 1 Report

Comments and Suggestions for Authors

I am satisfied with the author's comments except for one in which the author has not provided p-values for regression analysis.

Kindly provide the p-values for regression analysis 

Author Response

Dear Reviewer, Thank you very much for taking the time to review this article. Specific suggestions and comments regarding the "Submission ID: healthcare-3890681" are listed below. Each change has been made point by point, and corrections and revisions are included in the main manuscript. Thank you for your attention. Best regards Respond 1#. I am satisfied with the author's comments except for one in which the author has not provided p-values for regression analysis. Kindly provide the p-values for regression analysis. Thank you very much for this justified warning. The existing data sources were re-analyzed to determine the p-values and added to Table 3. All corrections are highlighted in yellow within the main text. Thank you very much for your contributions.

Reviewer 3 Report

Comments and Suggestions for Authors

The manuscript is generally well-organized and clearly written. The study addresses an important public health topic related to rational drug use and patient behavior in primary care settings. The methodology is appropriate for the study objectives, and the results are presented clearly. The discussion effectively links the findings to previous research and highlights relevant implications for public health practice.

Only minor editorial improvements are recommended, such as light language polishing and ensuring consistency in references and figure/table labeling. Overall, the paper has been substantially improved from the previous version and is suitable for publication in its current form or with minimal revisions.

  • Tables and Figures: Ensure that all captions and abbreviations are consistent with those used in the text.

  • Discussion: Briefly expand on the possible explanation for the negative correlation between health literacy and medication adherence.

Author Response

Dear Reviewer, Thank you very much for taking the time to review this article. Specific suggestions and comments regarding the "Submission ID: healthcare-3890681" are listed below. Each change has been made point by point, and corrections and revisions are included in the main manuscript.

Thank you for your attention.

Sincerely

Respond 1#.The manuscript is generally well-organized and clearly written. The study addresses an important public health topic related to rational drug use and patient behavior in primary care settings. The methodology is appropriate for the study objectives, and the results are presented clearly. The discussion effectively links the findings to previous research and highlights relevant implications for public health practice. Only minor editorial improvements are recommended, such as light language polishing and ensuring consistency in references and figure/table labeling. Overall, the paper has been substantially improved from the previous version and is suitable for publication in its current form or with minimal revisions. Tables and Figures: Ensure that all captions and abbreviations are consistent with those used in the text. Discussion: Briefly expand on the possible explanation for the negative correlation between health literacy and medication adherence.

Thank you very much for your kind comments and constructive criticism. These suggestions are really important to increase the impact of our article. The manuscript has been reviewed, and necessary language improvements have been made. Inconsistencies in reference and figure/table labeling have been corrected. Tables and Figures: All captions and abbreviations were re-checked in the text and necessary edits were made. Discussion: Studies in the literature indicate significant differences in health literacy, even among well-educated groups such as university students (Chen et al., Healthcare 2023; Persell et al., Patient Preference and Adherence 2020). A high level of formal education does not automatically translate to high health literacy, as individuals may lack the critical skills needed to effectively evaluate and apply health information. High health literacy does not guarantee high medication adherence. Indeed, the correlation analysis in our study also revealed this result. Similar to the paradox in our study, a negative correlation was reported between health literacy and adherence to medication treatment in the study conducted by Persell et al. (Persell et al., Patient Preference and Adherence 2020). Therefore, health communication strategies, both in universities and in the broader community, should prioritize improving health literacy. This statement was also added to the final part of the discussion. All corrections are highlighted in yellow within the main text.

Thank you very much for your contributions.
